PERSPECTIVE

# Focusing on cognitive potential as the bright side of mental atypicality

Lorenza S. Colzato[1,2,3], Christian Beste [1,2,3,4✉] & Bernhard Hommel [1,2,3,4]

Standard accounts of mental health are based on a "deficit view" solely focusing on cognitive impairments associated with psychiatric conditions. Based on the principle of neural competition, we suggest an alternative. Rather than focusing on deficits, we should focus on the cognitive potential that selective dysfunctions might bring with them. Our approach is based on two steps: the identification of the potential (i.e., of neural systems that might have benefited from reduced competition) and the development of corresponding training methods, using the *testing-the-limits* approach. Counterintuitively, we suggest to train not only the impaired function but on the function that might have benefitted or that may benefit from the lesser neural competition of the dysfunctional system.

Psychiatry and clinical psychology, except humanistic and positive psychology based on encouraging human potential[1–3], are driven by a deficit view: people whose mental performance deviates to a particular (often not well-defined and justified) degree from what is considered average human performance are deemed to be "ill" and in need of correction—with the therapeutic aim of reducing the discrepancy between their performance and the population mean. The deficit view is shared by many other disciplines, including linguistics, sociology, education, disability studies, and anthropology, which all try to characterize atypical individuals in terms of their observed deficiencies, dysfunctions, difficulties, challenges, and limitations[4–7]. This deficit view has been criticized because it stigmatizes both atypical behaviors and the people showing them, with potentially severe negative personal and social consequences[5,8–11]. In the field of anthropology, Taylor and colleagues[12] coined the term *complementary cognition* to point out that human cognitive evolution is likely to have resulted in individuals specialized in different but complementary neurocognitive search strategies (i.e., exploratory or exploitative activities). In so doing, human evolution has created a balance in individual neurocognitive specialization to enable an efficient adaptation to guarantee the survival of the human species[12]. Somewhat along these lines, researchers in the field of autism research have called for a shift from deficit-based to abilities-focused approaches to counteract the over-pathologization of human differences[5,13]. However, a systematic theoretical framework to provide the needed guidance for such a shift is lacking, so that the societal problems of the deficit view remain much better defined than possible alternatives.

## Implications of neural competition

Here we would like to suggest a possible avenue towards such a theoretical framework. An avenue that does not deny deficits or at least underperformance as compared to some standards, but that also considers the possible positive potential that such deficits or challenges might point to. Pretty much like large shadows imply a vital source of light, mental weaknesses might point to mental potential, at least in many cases, even though this potential would often need to be identified and systematically developed. Our current approach was stimulated by our recent

[1] Cognitive Neurophysiology, Department of Child and Adolescent Psychiatry, Faculty of Medicine, TU Dresden, Dresden, Germany. [2] Cognitive Psychology, Faculty of Psychology, Shandong Normal University, Jinan, China. [3] University Neuropsychology Center, Faculty of Medicine, TU Dresden, Dresden, Germany. [4]These authors contributed equally: Christian Beste, Bernhard Hommel. ✉email: christian.beste@uniklinikum-dresden.de

considerations on the downsides of cognitive enhancement[14], and it can be viewed as the flipside of these considerations. One of the two principles that our previous article was based on is the *principle of neural competition*. It refers to the fact that the human brain is capacity-limited[15–17] and that one of its essential characteristics is that neurons and neural networks compete for the representation and processing of environmental and internal information[18–20], as indicated in Fig. 1a. If so, strengthening one particular function or system or representational space through cognitive enhancement training would be expected to impair other processes or systems or representational spaces, as indicated in Fig. 1b. Indeed, studies employing transcranial direct current stimulation (tDCS), a noninvasive brain stimulation technique, have revealed a trade-off between enhanced and non-enhanced cognitive functions via the modulation of the cortical excitation/inhibition balance in the stimulated brain area[21–23]. Hence, stimulating the brain via tDCS to attain cognitive enhancement can increase one function but at the expense of another one[14]. In line with this idea, it has been shown that cognitive enhancement and impairment can be obtained within the same stimulation protocol[24]: Stimulating the dorsolateral prefrontal cortex (DLPFC) impairs learning while increasing automaticity for the learned material[24] and stimulating the posterior parietal cortex (PPC) facilitates learning while hampering automaticity[24]. Given that enhancement studies commonly focus on the to-be-enhanced function, these impairments are likely to go unnoticed, and we suggested spending more attention on possible costs of enhancement[14].

Interestingly, however, this view has a flipside that is indicated in Fig. 1c. Namely, if a particular function or system or representational space is less well-functioning, less well-developed in a given individual, as is suspected from neurologically and psychiatrically atypical individuals, this function/system/representational space should be a less potent competitor in the human brain. If thus, other functions/system/representational spaces are facing less competition, they either should have taken the opportunity to develop more efficiently than they would have with stronger neural competition, or they should at least have the potential to develop further. Evidence supporting this expectation is available from various lines of neurocognitive research, like obvious from the following three examples.

First, studies on blind individuals have indicated enhanced potential in auditory, tactile, and other kinds of perception, presumably as a direct consequence of the lack of use of what in sighted individuals are considered "visual" areas through/for visual perception[25–27]. To counterbalance the loss of a sense, the human brain generates or builds up corticocortical or sub-corticocortical connections between the deprived and the intact senses[28,29], suggesting that cross-modal plasticity is an adaptive process that enhances the remaining senses in blind people[30]. The visual cortex of blind people often exhibits a functional and structural reorganization, as indicated by activation of "visual areas" during odor detection, categorization, and discrimination[31,32] and tactile perception[33]. Findings of this sort point to extensive cross-modal plasticity[25–27], with regions of blind people's visual becoming increasingly responsive to input from other sensory channels. Consistent with this idea, the stimulation of the visual cortex via transcranial magnetic stimulation (TMS) in blind participants did not evoke phosphenes but elicited tactile sensations[34]. That is, lack of competition from the visual modality for representational space in the visual cortex creates the potential to enhance processing through other sensory modalities. Indeed, compensatory plasticity, in many cases, leads to the development of supranormal skills when using one of the remaining senses, such as improved tactile grating orientation and better pitch discrimination in blind people compared to sighted controls[26].

Second, studies on phantom limb syndrome show that amputees still experience sensations from the amputated limb[35]. Similar to blindness, these cases reflect a cortical reorganization[36–38], which makes neural representational spaces that had previously been used by the now amputated limb available for the representations of other body parts, commonly those with adjacent representations in somatosensory and motor maps[39,40]. Consistent with neuroimaging evidence, a TMS study, indexing cortical mapping before and after upper limb amputation, showed that the neighboring areas took over the deafferented zone[41]. Consequently, the amputee might attribute signals resulting from movements of these body parts as indicating movement of the amputated limb, which violates expectations and is often experienced as pain. Indeed, it has been shown that the degree of the somatotopic shift from the lip map to the deafferented hand map predicts the severity of phantom pain[42]. That is, when the brain's primary sensorimotor cortex no longer gets inputs from the amputated hand, signals from the lips begin to take over that area. Such a maladaptive reorganization of the sensorimotor cortex is likely to cause pain in the phantom limb, as absence of inhibitory activity in the sensory-cortical feedback pathways triggers continued efferent motor cortical commands as a result of enhanced cortical excitability[43].

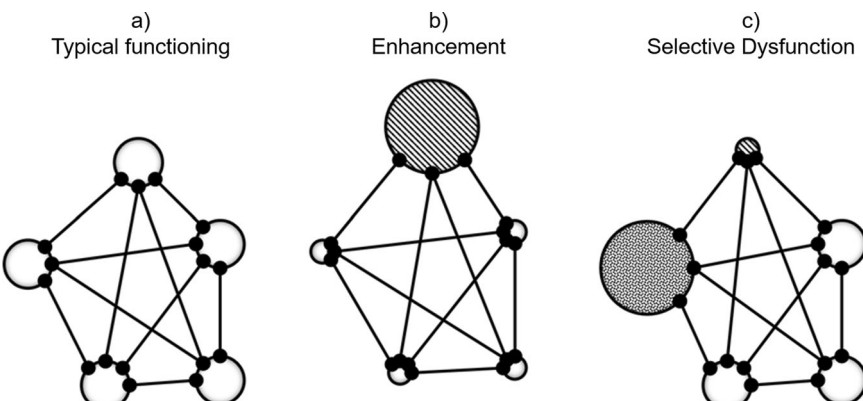

**Fig. 1 Consequences of enhancement and selective dysfunction in a competitive neural system. a** Sketch of neural brain systems (indicated by white circles) in a network characterized by mutually competitive competition (indicated by mutually inhibitory connections, see lines and black circles). **b** Consequences of selective cognitive enhancement: enhancing a system (in terms of its functioning and/or representational space) is likely to reduce the efficiency of other systems in the competitive network. **c** Hypothesized consequences of selective dysfunction of one system: another system or other systems might benefit from the lack of competition coming from the dysfunctional system.

Third, ocular dominance has been shown to result from the competition between neurons and neural networks to represent and process environmental and internal information[44–46]. For instance, the dominance of one eye over the other was a direct function of the amount of light stimulating the dominant eye in birds[44]. Further, an individual increase of right-eye dominance substantially enhanced the grain–grit discrimination success in pigeons by 10%, suggesting that high discrimination accuracy is mainly related to right-eye dominance[45]. This suggests that neural competition during neurobiological development is driven by the active use of the competing systems, with more use of one system reducing the functioning of others.

**A cognitive-potential approach**. The principle of neural competition suggests that dysfunctional neural systems are likely to be weaker competitors in the neural competition, which might be beneficial for competitors. Of course, one could imagine scenarios in which no beneficiary exists, perhaps because the cortical development of a natural beneficiary was completed before competition from the less efficient system was sufficiently reduced, probably because the dysfunctional nature of the less efficient system created more disturbing "cortical noise" than the benefit from reduced competition could have helped, or because the extra capacity or representational space offered by the less efficient system was of no practical use for any other system. Hence, we do not claim that the low efficiency of one system *must* lead to more efficiency in another—all we claim is that this is a possibility. Moreover, it may well be that low efficiency in one system has already boosted the functionality of another. For instance, damage to the left cortical hemisphere and the language-related areas it houses in most adults relatively early in the ontogenetic development leads to a relatively efficient take-over from the developing right hemisphere, which can lead to the entirely unimpaired acquisition of language[47]. In this case, the brain may already have exploited available resources to an optimal degree, so further optimization attempts might very well lead to overall impairments. For instance, the principle of neural competition is not unlikely to rely on changes in neurotransmitter levels[19], which are known to relate to cognitive performance in a nonmonotonic, often curvilinear (inverted-U) relationship[48], so that further pushing a system or function towards better functioning might actually impair performance.

More interesting for our approach are cases in which the potential that the lower efficiency of one system might provide has not yet been already identified and actively used by other systems. Substantial and systematically guided training might be necessary to use the full potential of unused cortical capacities. Accordingly, our cognitive-potential approach calls for two successive steps that fully exploiting the cognitive potential of individuals must entail. Before describing these two steps in turn, we would like to emphasize that developing possible potential may not be without risk. On the one hand, the situation sketched in Fig. 1 might represent the endpoint of some already completed ontogenetic process, which would imply that the degree of dysfunctionality of the dysfunctional system is relatively permanent. In this case, attempts to develop remaining potentials of other systems would not be risky. On the other hand, however, the process leading to the dysfunctionality might be still underway. If so, developing the potential of other systems might increase the competition with the dysfunctional system and further impair its functionality. Obviously, in these cases attempts to maintain the present level of (dys)functionality of the impaired system would have absolute priority over other cognitive-enhancement strategies.

**Step 1: Identifying beneficiaries of selective dysfunction**. As a first step, it is important to assess the individual potential of individuals with neural systems that are considered dysfunctional to some degree. This requires the identification of those neural systems that might have benefited or that might still benefit from the lesser neural competition from the dysfunctional system. Even though there is no established theoretical framework that could provide a list of possible candidates, the available evidence suggests at least three criteria that are likely to provide a promising search template. First, studies indicating reorganization in amputees suggest that neuroanatomical proximity can be an important cue[36,39,40]. Hence, representational spaces or functions closest to a dysfunctional system might be suspected to have taken over parts of the neural capacity that a more functional system would have occupied. Second, studies on blind individuals and on the neural development in pigeons[26,44,45] suggest that functional alternatives can benefit from lesser competition: impaired functioning of one sensory modality is likely to benefit other modalities, impaired functioning of one symmetrical system, like the eye or a cortical hemisphere, is likely to benefit the other one. Third, evidence on the antagonistic nature of some neural systems suggests that weaknesses of one competitor might benefit its natural competitor. For instance, behavioral and neurocognitive studies have suggested that dopaminergic systems in prefrontal cortex might interact antagonistically with dopaminergic systems in the striatum to bias information processing towards persistence (high selectivity, focus on currently relevant information only) or flexibility (integrative, parallel processing), respectively[49–51]. Among other things, this suggests that dysfunctions related to prefrontal cortex might impair cognitive functions relying on this area but promote functions relying on the striatum—i.e., prefrontal dysfunction might promote flexibility. There is indeed some evidence in favor of this prediction. In individuals diagnosed with OCD and in individuals diagnosed with ADHD, who both suffer from an unbalance of dopaminergic systems in prefrontal cortex and in the striatum[52,53], some cognitive functions have been reported to be impaired as compared with healthy matched controls, whereas others are apparently improved. We suggest that this paradox can be resolved if OCD and ADHD are not considered independent categories but as pointers to the two opposite poles of a common dimension, very much along the lines of the antagonistic persistence-flexibility dimension[54,55]. The potential to disentangle what is signal and what is noise represents the basis for a successful balance between the antagonistic poles of the persistence-flexibility dimension. During information processing, the level of a desired signal to the level of background neuronal noise, also called signal-to-noise ratio (SNR)[56–60], is the most conceivable neural candidate underpinning the antagonistic persistence-flexibility dimension, also referred to as the metacontrol hypothesis of cognitive control[54,55]: less noise (i.e., high SNR) implies a more stable cognitive state[61–63], while more noise (i.e., low SNR) might generate more behavioral variability[64,65], implying a flexible cognitive state. Following these lines of reasoning, the trade-off between a functional and a dysfunctional system depends on the SNR: high SNR might support a more stable cognitive state (i.e., cognitive persistence), but at the costs of variable cognitive state (i.e., cognitive flexibility), and the opposite holds for low SNR.

Indeed, OCD has been considered an impairment in cognitive flexibility (Gruner and Pittenger, 2017), which fits with the evidence of an altered SNR as indexed by sensorimotor gating[66,67] and with the well-known phenomenology of repetitive behavior with rigid rituals and diminished behavioral flexibility[68]. At the same time, individuals diagnosed with OCD were reported to outperform healthy controls in tasks requiring a focused state[69]

or in tasks requiring the selective reactivation of a previously inhibited mental set[70]. Hence, OCD seems to be characterized by atypically bad performance in flexibility-heavy tasks and atypically good performance in persistence-heavy tasks. Conversely, individuals diagnosed with ADHD display low SNR as measured by "1/*f noise*", index of scale-free neural activity in EEG, which resulted to be increased after methylphenidate use[57], and deficits in tasks requiring a focused cognitive state, such as sustained attention and vigilance[71], but they outperform healthy controls in divergent thinking, which requires the generation of many different ideas[72], in exploratory foraging patterns[73], and in the implicit learning of an artificial grammar[74]. Hence, ADHD seems to be characterized by atypically bad performance in persistence-heavy tasks and atypically good performance in flexibility-heavy tasks.

It is important to consider that the available findings might not generalize to each single individual. For instance, particular genetic predispositions might prevent or hamper the occupation of underused cortical capacities for other purposes, or the cortical organization might be such that antagonistic relationships between systems might not have emerged as in more typical brains. Accordingly, the identification of potential can certainly be guided by available findings, which often rely on sample means, but eventually needs to be tailored to each individual.

**Step 2: Training to the limit**. Once the potential has been identified, the question will be how to make use of it. One important moderator for this question will be individual differences. Obviously, some individuals might have identified their corresponding potentials already and are already experienced in making optimal use of it. Others may have focused on their deficits only and may not have spent any efforts on developing their potential. Even others may not even have a particular potential, for reasons as mentioned in the previous section. Accordingly, the second step needs to be based on individual assessments of a given person's potential. Given that the degree to which a given individual has already exploited her potential will often be hard to determine objectively, we recommend to apply the logic of the *testing-the-limits* research approach proposed by Baltes and colleagues to study performance in the elderly[75–77]. According to this approach, simply comparing cognitive performance across particular groups that are assumed to differ makes little sense, because the causes underlying possible significant differences remain uncertain: it might be a lack of capacity, a lack of using the capacity, or both. To determine true capacity limitations, it is not essential to know what the current spontaneous performance is but, rather, what the optimal performance can be. In other words, the approach is interested in potential. To assess true potential, training methods need to be identified to optimize the use of this potential, so to see which levels of performance can be reached after this training. Hence, in line with this *testing-the-limits* approach, we call for "selective optimization with compensation"[75–79] which assumes that the best way to maintain high levels of cognitive performance is to focus and restrict the training on the potentially enhanced domain of functioning unaffected by the diagnosed condition. With respect to our OCD-ADHD example, individuals diagnosed with OCD should be selectively trained in persistence while individuals diagnosed with ADHD should be selectively trained in flexibility. Note that this approach suggests the exact opposite of what can be considered the more intuitive and widely used therapeutic strategy: while intuition would suggest training people in what they are bad at, our approach suggests training people in what they could become good at. Strengthening potential rather than trying to repair weaknesses is not only likely to be more efficient, but it can also

be expected to be more motivating for the diagnosed individuals and to foster their self-respect—the lack of which is indeed a common problem in atypical individuals[80–84].

**Outlook**. Standard accounts of mental health are based on a "deficit view" solely focusing on cognitive impairments associated with psychiatric conditions. This view is known to lead to personal discouragement, lack of self-respect, and societal stigmatization, which often further increase the problems diagnosed individuals are facing. The "deficit view" has been challenged also in the field of anthropology by the idea of the evolution of complementary cognition, suggesting that successful adaptation arises from the cooperation of individual members who are neurocognitively specialized in different but complementary neurocognitive search strategies[12]. According to complementary cognition, once a certain threshold has been reached, the only efficient way to increase brain capacity is via specialization. As a result of the trade-off between a functional and a dysfunctional complementary system, the specialization in one system comes at the cost of an impairment in the complementary system[12]. Based on the principle of neural competition, in line with the idea of the evolution of complementary cognition, we suggest an alternative to the "deficit view": rather than focusing on deficits, we should focus on the possible cognitive potential that selective dysfunctions might bring with them. Our approach is based on two steps: the identification of the potential (i.e., of neural systems that might have benefited from reduced competition already or that might benefit after specific training) and the development of corresponding training methods, using the *testing-the-limits* approach. Counterintuitively, we thus suggest to target training not on the impaired function but, rather, on the function that might have benefitted or that may still benefit from the lesser neural competition of the dysfunctional system. This approach of identifying potential mental gains and their possible magnification is likely to promote educational achievement and the development of self-esteem in diagnosed individuals.

**Reporting summary**. Further information on research design is available in the Nature Research Reporting Summary linked to this article.

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

## Acknowledgements
This work was supported by Grants from the Deutsche Forschungsgemeinschaft, SFB 940, SFB TRR 265 and FOR 2698.

## Author contributions
L.S.C., C.B. and B.H. contributed equally to writing and finalizing the manuscript.

## Funding

## Competing interests
The authors declare no competing interests. C.B. is an Editorial Board Member for *Communications Biology*, but was not involved in the editorial review of, nor the decision to publish this article.
