## [Peer Review File · Communications Biology]

Reviewers' comments:

Reviewer #1 (Remarks to the Author):

This is a fantastic article, using simple straightforward examples to lay a foundation for future enquiry, in a field which has heretofore lacked theoretical framing. Well done.

I recommend that you also review the work of Dr Helen Taylor which speaks to how such variance in human neurotypes may have evolved and for what species-level purpose. There's complementary synergy in the propositions.

Taylor, H., Fernandes, B., & Wraight, S. (2021). The Evolution of Complementary Cognition: Humans Cooperatively Adapt and Evolve through a System of Collective Cognitive Search. *Cambridge Archaeological Journal*, 1–17. <https://doi.org/10.1017/s095977432100032>

I recommend that you add a few sentences on Complementary Cognition in order to connect the individual level of analysis to the species level. Bringing the two theoretical frameworks together in one place will benefit of future research, which will no doubt circle back to this as a key paper.

Reviewer #2 (Remarks to the Author):

Review of "THE BRIGHT SIDE OF MENTAL ATYPICALITY: A NEUROCOGNITIVE POTENTIAL APPROACH" by Colzato, Beste, and Hommel

Manuscript Overview: The authors suggest that there are potentially a wide range of neurological deficits that are part of "push-pull" systems (i.e., where the mechanism that produces weaknesses in one type of mental function, at the same time produces advantages in another). The authors further argue that while the standard viewpoint in the literature is one that is focused on the deficit and how to reduce it, an alternative approach may be to try to augment the counter-point ability.

Review Overview: Overall, I think it's an interesting and provocative piece. It definitely made me think really deeply about a number of issues that I hadn't considered previously.

I really have just one main comment/issue (that's somewhat split into two halves below).

1) Although there's potentially interesting theory in here, it seems that really the primary focus is on what might "do the most good" in an applied sense.

Given the applied goal then, it seems like a pretty critical step is missing – which is to first evaluate whether more good will be done by trying to improve the deficient process or the advantaged process. This is particularly true since, at least based upon the theory put forward, improving the advantaged process should, at the same time, further degrade the deficient process.

Let's say for instance, that an individual has low vision (e.g., far worse than normal 20/20 vision, but sufficiently preserved that the input can be used for some types of navigation in the real-world). The authors state that, "the best way to maintain high levels of cognitive performance is to focus and restrict the training on the potentially enhanced domain of functioning unaffected by the diagnosed condition." In this case that might be the auditory system. But if training the auditory system caused what visual abilities still remained to be eliminated, that feels like a net loss?

Similarly, let's say there's an individual who has just enough sustained attention that they can make it through a college class. If, by training them on cognitive flexibility, their sustained attention abilities are reduced to the point that they can no longer make it through a class, that also feels like a net loss?

So, I feel like some further elaboration of these ideas is necessary – whether it's explaining why enhancing the unaffected process won't lead to further degradation of the deficient process (it seems like it would have to if the processes were truly push-pull; e.g., if they were based on

balance of excitation/inhibition, or dopamine balance, or upon competition for neural representation), or else by better describing the decision-making topography (e.g., when/under what circumstances might further declines in the deficient process be "worth it" given the gains in the unaffected process?).

2) And then, on the "flip side" of the above, the other issue that needs to be engaged with are possible harms that could go along with further pushing along an already advantaged process. For instance, let's say an individual has unusually high neural excitability (e.g., related to glutamate or acetylcholine) and that leads to worse performance on tasks that require more inhibition. It seems potentially problematic in that case to continue to push toward even more excitability (i.e., because doing so could de-stabilize the entire network).

So again, there would be virtue in some elaboration on how to decide when to push forward an already advantaged process (i.e., in order for it to be a good idea there'd need to be a monotonic relationship between pushing on that advantaged process and outcomes getting ever better; the relationship couldn't be curvilinear where, after some level, things get overall way worse). It seems like those types of curvilinear relationships are almost necessarily always going to be found given that we're working with a biological substrate (e.g., if being strong is good, being even stronger is better...until you're so strong that you can apply more force than your tendons/ligaments can take).

IN DETAIL:

Reviewers' comments:

Reviewer #1:

This is a fantastic article, using simple straightforward examples to lay a foundation for future enquiry, in a field which has heretofore lacked theoretical framing. Well done.

REPLY: We thank the reviewer for the positive appreciation of our manuscript.

I recommend that you also review the work of Dr Helen Taylor which speaks to how such variance in human neurotypes may have evolved and for what species-level purpose. There's complementary synergy in the propositions.

Taylor, H., Fernandes, B., & Wraight, S. (2021). The Evolution of Complementary Cognition: Humans Cooperatively Adapt and Evolve through a System of Collective Cognitive Search. Cambridge Archaeological Journal, 1–17. <https://doi.org/10.1017/s095977432100032>

I recommend that you add a few sentences on Complementary Cognition in order to connect the individual level of analysis to the species level. Bringing the two theoretical frameworks together in one place will benefit of future research, which will no doubt circle back to this as a key paper.

REPLY: We thank the reviewer for pointing out the work by Dr. Taylor that we are happy to discuss and cite. We included the following new text in the introduction:

“In the field of anthropology, Taylor and colleagues¹² coined the term complementary cognition to point out that human cognitive evolution is likely to have resulted in individuals specialized in different but complementary neurocognitive search strategies (i.e., exploratory or exploitative activities). In so doing, human evolution has created a balance in individual neurocognitive specialization to enable an efficient adaptation to guarantee the survival of the human species¹².”

And in the conclusion:

“The “deficit view” has been challenged also in the field of anthropology by the idea of the evolution of complementary cognition, suggesting that successful adaptation arises from the cooperation of individual members who are neurocognitively specialized in different but complementary neurocognitive search strategies¹². According to complementary cognition, once a certain threshold has been reached, the only efficient way to increase brain capacity is via specialization. As a result of the trade-off between a functional and a dysfunctional complementary system, the specialization in one system comes at the cost of an impairment in the complementary system¹².”

Reviewer #2:

Manuscript Overview: The authors suggest that there are potentially a wide range of neurological deficits that are part of “push-pull” systems (i.e., where the mechanism that produces weaknesses in one type of mental function, at the same time produces advantages in another). The authors further argue that while the standard viewpoint in the literature is one that is focused on the deficit and how to reduce it, an alternative approach may be to try to augment the counter-point ability.

Review Overview: Overall, I think it's an interesting and provocative piece. It definitely made me think really deeply about a number of issues that I hadn't considered previously.

REPLY: We thank the reviewer for the positive appreciation of our manuscript.

I really have just one main comment/issue (that's somewhat split into two halves below).

1) Although there's potentially interesting theory in here, it seems that really the primary focus is on what might "do the most good" in an applied sense.

Given the applied goal then, it seems like a pretty critical step is missing – which is to first evaluate whether more good will be done by trying to improve the deficient process or the advantaged process. This is particularly true since, at least based upon the theory put forward, improving the advantaged process should, at the same time, further degrade the deficient process.

Let's say for instance, that an individual has low vision (e.g., far worse than normal 20/20 vision, but sufficiently preserved that the input can be used for some types of navigation in the real-world). The authors state that, "the best way to maintain high levels of cognitive performance is to focus and restrict the training on the potentially enhanced domain of functioning unaffected by the diagnosed condition." In this case that might be the auditory system. But if training the auditory system caused what visual abilities still remained to be eliminated, that feels like a net loss?

Similarly, let's say there's an individual who has just enough sustained attention that they can make it through a college class. If, by training them on cognitive flexibility, their sustained attention abilities are reduced to the point that they can no longer make it through a class, that also feels like a net loss?

So, I feel like some further elaboration of these ideas is necessary – whether it's explaining why enhancing the unaffected process won't lead to further degradation of the deficient process (it seems like it would have to if the processes were truly push-pull; e.g., if they were based on balance of excitation/inhibition, or dopamine balance, or upon competition for neural representation), or else by better describing the decision-making topography (e.g., when/under what circumstances might further declines in the deficient process be "worth it" given the gains in the unaffected process?).

REPLY: Very interesting consideration. We openly discuss this possibility relating this point with the later point raised by reviewer 2. We included this text:

"Moreover, it may well be that low efficiency in one system has already boosted the functionality of another. For instance, damage to the left cortical hemisphere and the language-related areas it houses in most adults relatively early in the ontogenetic development leads to a relative efficient take-over from the developing right hemisphere, which can lead to entirely unimpaired acquisition of language⁴⁷. In this case, the brain may already have exploited available resources to an optimal degree, so that further optimization attempts might very well lead to overall impairments. For instance, the principle of neural competition is not unlikely to rely on changes in neurotransmitter levels¹⁹, which are known to relate to cognitive performance in a nonmonotonic, often curvilinear (inverted-U) relationship⁴⁸, so that further pushing a system or function towards better functioning might actually impair performance.

More interesting for our approach are cases in which the potential that the lower efficiency of one system might provide has not yet been already identified and actively used by other systems. Substantial and systematically guided training might be necessary to use the full potential of unused cortical capacities. Accordingly, our cognitive-potential approach calls for two successive steps that fully exploiting the cognitive potential of individuals must entail. Before describing these two steps in turn, we would like to emphasize that developing possible potential may not be without risk. On the one hand, the situation sketched in Figure 1 might represent the endpoint of some already completed ontogenetic process, which would imply that the degree of dysfunctionality of the dysfunctional system is relatively permanent. In this case, attempts to develop remaining potentials of other systems would not be risky. On the other hand, however, the process leading to the dysfunctionality might be still underway. If so, developing the potential of other systems might increase the competition with the dysfunctional system and further impair its functionality. Obviously, in these cases attempts to maintain the present level of (dys)functionality of the impaired system would have absolute priority over other cognitive-enhancement strategies.”

2) And then, on the “flip side” of the above, the other issue that needs to be engaged with are possible harms that could go along with further pushing along an already advantaged process. For instance, let’s say an individual has unusually high neural excitability (e.g., related to glutamate or acetylcholine) and that leads to worse performance on tasks that require more inhibition. It seems potentially problematic in that case to continue to push toward even more excitability (i.e., because doing so could de-stabilize the entire network).

So again, there would be virtue in some elaboration on how to decide when to push forward an already advantaged process (i.e., in order for it to be a good idea there’d need to be a monotonic relationship between pushing on that advantaged process and outcomes getting ever better; the relationship couldn’t be curvilinear where, after some level, things get overall way worse). It seems like those types of curvilinear relationships are almost necessarily always going to be found given that we’re working with a biological substrate (e.g., if being strong is good, being even stronger is better...until you’re so strong that you can apply more force than your tendons/ligaments can take).

REPLY: We completely agree with the line of reasoning suggested by reviewer 2. Accordingly, we merged the two points raised by the reviewer in the added text outlined above.

REVIEWERS' COMMENTS:

Reviewer #1 (Remarks to the Author):

Well done again, this addition fits in well and addresses my point.

I have no further edits and look forward to publication of this paper.

Reviewer #2 (Remarks to the Author):

The section that the authors inserted in response to my two (linked) comments does a perfect job of addressing the minor issues I had. I have no further questions/critiques. As I noted in my previous review, I think the ideas are provocative and should stir some really interesting research lines.

Reviewer comments

There are no further reviewer comments left. We thank the unknown reviewers for the constructive feedback.